# Behind-The-Scenes (BTS): Wiper-Occlusion Canceling for Advanced Driver Assistance Systems in Adverse Rain Environments

**DOI:** 10.3390/s21238081

**Published:** 2021-12-02

**Authors:** Junekyo Jhung, Shiho Kim

**Affiliations:** Seamless Transportation Lab (STL), School of Integrated Technology, Yonsei Institute of Convergence Technology, Yonsei University, Incheon 21983, Korea; ejhung@yonsei.ac.kr

**Keywords:** windshield wiper, object detection, optical flow, data synthesis, advanced driver assistance systems, adverse rain

## Abstract

Driving in an adverse rain environment is a crucial challenge for vision-based advanced driver assistance systems (ADAS) in the automotive industry. The vehicle windshield wiper removes adherent raindrops that cause distorted images from in-vehicle frontal view cameras, but, additionally, it causes an occlusion that can hinder visibility at the same time. The wiper-occlusion causes erroneous judgments by vision-based applications and endangers safety. This study proposes behind-the-scenes (BTS) that detects and removes wiper-occlusion in real-time image inputs under rainy weather conditions. The pixel-wise wiper masks are detected by high-pass filtering to predict the optical flow of a sequential image pair. We fine-tuned a deep learning-based optical flow model with a synthesized dataset, which was generated with pseudo-ground truth wiper masks and flows using auto-labeling with acquired real rainy images. A typical optical flow dataset with static synthetic objects is synthesized with real fast-moving objects to enhance data diversity. We annotated wiper masks and scenes as detection ground truths from the collected real images for evaluation. BTS outperforms by achieving a 0.962 SSIM and 91.6% F1 score in wiper mask detection and 88.3% F1 score in wiper image detection. Consequently, BTS enhanced the performance of vision-based image restoration and object detection applications by canceling occlusions and demonstrated it potential role in improving ADAS under rainy weather conditions.

## 1. Introduction

Recently, the automotive industry has focused on the implementation of autonomous vehicles mounted with only cameras, which, like human drivers, primarily depend on visual perception [1,2], as image data have proven to be the richest source of raw data for advanced driver assistance systems (ADAS) to automate driving tasks through significant advancements in vision-based deep learning methods. One of the autonomous driving market-leading companies, Tesla’s Autopilot, relies primarily on cameras to perceive its environment without radar and LiDAR sensors [3]. As a result, adaptive cruise control (ACC), automatic emergency braking (AEB), lane keeping assistance (LKA), and many other ADAS tasks leverage vision-based tasks, such as object detection, classification, tracking, and depth estimation [4]. However, vehicles generally experience meteorological phenomena, such as rain, so visual data from mounted cameras often suffer quality degradation because of rain streaks, rain accumulation, or adherent raindrops. In terms of images, rain streaks generate bright straight lines that partially occlude the foreground and background. Rain accumulation comprises multiple layers of rain streaks and creates effects similar to fog or haze that blur the image content. Adherent raindrops are created directly on a camera lens or vehicle windshield and distort images with relatively larger regions than other rain effects. Recently, Tesla also addressed the issue of error-causing debris in images in a hardware manner [5]. Failure in recognition or detection resulting from impaired images may cause erroneous decisions or control, which is dangerous in terms of safety. Therefore, coping with the effects of rainy weather conditions is indispensable for enhancing vision-based systems in vehicles.

Deraining [6,7,8,9,10,11,12,13,14,15,16], a software method to remove rain effects, have shown promising research results, but there are constraints in applying them to real environments because of gaps in rain effects. Most approaches are trained based on unrealistic synthetic or synthesized datasets because obtaining precise ground truth data regarding relatively small rain objects is a difficult process. Empirically, the actual impacts of rain effects on driving can vary, as illustrated in Figure 1. Rain streaks are barely visible in the captured images, and rain accumulation occurring at a distance has less influence on near objects. The experimental dataset shows that two of the rain effects are not crucial in actual images from driving under rainy weather conditions as much as deraining open datasets. Instead, for an adherent raindrop on a windshield, created in front of the in-vehicle camera, the spherical and transparent droplets cause severe image degradation, such as distortion in both open and experimental datasets. Modeling realistic synthetic adherent raindrops is complicated because the transparent fluid interacts with the surroundings with unpredictable patterns. Therefore, these intractable properties make the adherent object a difficult challenge for clarifying image visibility.

Although vision-based deraining approaches struggle with troublesome objects, vehicles have removed rain simply with a physical device, a windshield wiper, since 1903. This is one of the reasons why frontal view cameras for ADAS are integrated interior of vehicles to avoid ADAS performance degradation by the debris under rainy weather conditions. Nevertheless, although this intuitive and robust device brings clear visibility to an image, it creates unintentional occlusions, as shown in Figure 2. As the wiper operates upward and downward to remove adherent raindrops, it truncates or occludes objects in the image, such as vehicles and road features. If the frontal vehicle is completely occluded and never seen in sequential frames, a vision-based recognition application would misjudge the existence, location, size, or class of the object and cause faulty decisions in ADAS tasks. Likewise, if wiper-occlusions cause other vision-based applications, such as lane detection, drivable area detection, traffic sign detection, and many others, to output erroneous results to ADAS tasks, vehicles would make wrong decisions in lateral and longitudinal controls that endanger safety. As a result, the windshield wiper is the critical object that obstructs the visibility of camera sensors and causes fatal occlusions in images, and this should be resolved in driving environments.

Even though wipers occlude images and deteriorate vision-based performances, only one study has been conducted on removing the wiper [17]. In the past decade, vision-based deep learning methods for driving have demonstrated practical advancements by leveraging various open datasets. However, most datasets [18,19,20] capture images with ideal visibility under clear weather conditions, and a few datasets [21,22] include rainy scenes for data diversity, while windshield wipers have been excluded [23]. As previous studies have reported, employing real data for training and evaluation is crucial for implementing practical solutions compatible with actual driving environments. Therefore, it is important to obtain actual windshield wiper data to compensate for the absence of a dataset.

Auto-labeling is an approach for obtaining approximate data to improve the efficiency of annotation. For auto-labeling windshield wipers, general vision-based object detection [24,25], and segmentation [26] trained with wiper-free datasets are unsuitable because the wiper is a total stranger for these approaches, which produce impracticable and dissatisfactory results. In contrast, a state-of-the-art optical flow estimation network [27] outputs plausible results regarding an unseen object because the method infers by extracting not only object-wise features but also pixel-wise displacements of an image sequence. In other words, it is suitable for auto-labeling pixels related to the wiper, which have relatively larger displacements than other pixels. However, the network was trained with optical flow datasets [18,28,29,30] consisting of static synthetic images and flow data, and its ability with fast-moving wipers is inaccurate and unstable; therefore, improving the data diversity by adding dynamic objects of a real domain is required for precise windshield wiper detection to cancel occlusions in unseen data.

We propose a behind-the-scenes (BTS) for windshield wiper-occlusion canceling by leveraging optical flow to maintain clear visibility of images while driving under rainy weather conditions. The main purpose of BTS is to extract the mask of a windshield wiper, a relatively fast-moving object in an image, by exploiting the predicted optical flows from a sequential image pair. Our method facilitates the extension of data diversity in optical flow datasets, pixel-wise detection of actual windshield wipers, and restoration of wiper-occlusion regions, as depicted in Figure 3.

Real image data under rainy weather conditions were collected by driving a camera-mounted vehicle in a metropolitan city in Korea to address the absence of windshield wiper images. For data diversity, we acquired data considering various locations, precipitations, and seasons. Hand-crafted ground truth masks were generated only to evaluate the predicted wiper masks and scenes, while pseudo-ground truth masks and flows were extracted by auto-labeling using a pre-trained optical flow model. The idea of data synthesis is to mount a virtual wiper on a virtual camera lens of a synthetic dataset to implement intentional occlusions. Therefore, we overwrote the pseudo-data to MPI-Sintel [30], a commonly used optical flow dataset, by sampling data according to various synthesis scenarios.

We employed the RAFT [27] as a baseline to fine-tune the pre-trained weight of the network with our synthesized dataset containing diverse data. BTS takes two sequential images to infer pixel-wise optical flows, and a flow threshold classifies fast-moving wiper flows, similar to a high-pass filter. The structure similarity index measure (SSIM) and binary classification metrics were applied to evaluate the wiper mask detection (WMD) and wiper scene detection (WSD). Our contributions are as follows.

Acquisition of a real dataset of driving under adverse rainy weather conditions using windshield wipers;Implementation of a fine-tuning optical flow-based model with a synthesized dataset to detect precise windshield wiper-occlusion regions;Conception and realization of wiper-free rain images for autonomous driving datasets.

The remainder of this paper is organized as follows: Section 2 describes prior studies related to this research. Section 3 explains the methodology and processes for acquiring a new dataset and synthesizing the data. Section 4 reports the quantitative and qualitative evaluation results and demonstrates the extensibility and utility of vision-based applications. Section 5 discusses the results, limitations, and future work. We close this paper with the conclusions in Section 6. Video demonstration of BTS can be found in the Appendix A.

## 2. Related Work

### 2.1. Deraining

#### 2.1.1. Model-Driven Approaches

Until 2017, model-driven approaches were the prominent methods that rely on a statistical analysis of rain models and build a cost function to optimize. Kang et al. proposed a framework by formulating rain removal as an image decomposition into low/high-frequency parts based on morphological component analysis [6]. Luo et al. employed discriminative sparse coding [7], while Li et al. used patch priors based on Gaussian mixture models (GMM) to accommodate the orientations and scales of rain streaks [8]. You et al. exploited spatio-temporal derivatives of raindrops to remove adherent raindrops and restore the region by retrieving information from the image [9]. Despite these efforts, these approaches are inaccurate for real images because the actual rain shapes are inconsistent, especially in driving environments.

#### 2.1.2. Data-Driven Approaches

With advancements in deep learning, such as convolutional neural networks (CNNs) or generative adversarial networks (GANs), data-driven methods have begun to solve more complicated tasks and improve deraining performance. Yang et al. constructed a CNN-based multi-task deep learning architecture that progressively detects and removes rain streaks and rain accumulation [11]. Li et al. achieved significant performance improvement by preserving useful information in previous stages that can provide benefits in later stages for rain removal [12]. Zamir et al. and Chen et al. developed image restoration models that perform deraining more precisely than prior studies with respect to rain streaks and accumulation [13,14]. For adherent raindrops, Qian et al. employed a GAN by injecting visual attention into generative and discriminative networks to find, remove, and restore raindrop regions [15]. Liu et al. leveraged the motion differences between background and obstructing elements, i.e., raindrops, using optical flow and reconstructed decomposed background/obstruction layers [31]. However, these approaches utilize limited datasets in which images are different from real images.

#### 2.1.3. Deraining Datasets

Although deep learning-based approaches have enhanced deraining, performance degradation occurs when real rainy images are applied. In general, studies involving rain streaks and rain accumulation employ open datasets, such as Rain12 [8], Rain100L and Rain100H [11], Rain800 [16], Rain14000 [7], or MPID [32], in which synthetic and synthesized rain images occupy large portions because it is impossible to obtain rainy/clean image pairs from a natural scene. Furthermore, few adherent raindrop datasets exist for the same reason; therefore in [15], two pieces of glass, sprayed and clean, were used to acquire artificial real data. Nevertheless, unrealistic factors remain in the data, hindering their performance with respect to unseen real data.

### 2.2. Driving in Rainy Weather Conditions

#### 2.2.1. Deraining in Driving

Hnewa et al. employed various deraining models to real captured images under rainy conditions [33] from the BDD100K dataset to test how the models perform regarding domain mismatch. Then, they applied representative object detection methods, Faster R-CNN [34] and YOLO [24] trained with images under clear weather conditions, to the outputs from the deraining models. Their results demonstrate that deraining methods generate additional erroneous information in images, which degrades object detection performance.

As mentioned previously, Qian et al. removed adherent raindrops with a GAN [15], but its performance creates artifacts in our data as a result of domain mismatch. Fine-tuning may improve the performance, but it requires adherent raindrop mask ground truth data, which are difficult to obtain. Porav et al. made their stereo camera setup a bi-partite chamber with a clear acrylic panel in front of the lenses, with one section sprayed with water droplets using an internal nozzle, to create plausible adherent raindrop data [35]. Hirohashi et al. estimated the optical flows of occluded regions using a CNN rather than detecting raindrops [36]. Despite the efforts to generate realistic raindrops, the actual raindrops on a bent windshield of a moving vehicle have different features that these methods are unable to detect correctly.

#### 2.2.2. Wiper Removal

Adherent raindrops commonly appear on the windshield of an actual vehicle, and windshield wipers are mandatory devices to remove them, with apparent advantages and disadvantages. Although wipers clear adherent raindrops, they cause occlusion in the image data. Lin et al. classified wipers by training a principal component analysis (PCA) with extracted wiper masks from captured data using a constant camera position [17]. Dalal et al. suggested a hardware manner to remove actual wipers by replacing with pulsed laser to irradiate debris to clean a windshield [5]. To the best of our knowledge, our model is the first deep learning-based method to detect actual windshield wipers in real-time camera inputs; none of the previous methods aimed to detect the object. Although recent object detection [15,37,38] has shown excellent performance in driving environments, predicting a rectangular bounding box as a model for a large and bent wiper results in a great amount of non-target information in the box. Panoptic segmentation [26,39,40] classifies all pixels as independent objects but requires expensive efforts for annotation. Optical flow is another pixel-wise methodology applicable for moving object detection because it utilizes image sequences and [27] shows plausible adaptation to unseen data.

#### 2.2.3. Autonomous Driving Datasets

Open datasets have driven significant developments in vision-based autonomous driving tasks. The quality, quantity, and diversity of data from various sensor modalities contribute intuitively to related techniques. Most datasets [18,19,20,41,42] contain data under clear weather conditions, while some [21,22,43,44,45] provide data on driving in the rain. However, the distribution of rain data is small compared with other weather conditions. In addition, although most of them recorded image data through in-vehicle cameras, there are no images with windshield wipers because they were excluded [23]. Because of the absence of windshield wiper data, we acquired datasets using real road-driving experiments under inclement rain conditions.

### 2.3. Optical Flow

#### 2.3.1. Deep Learning-Based Approaches

Optical flow estimates the displacement vectors of all pixels in two images and is used for visual surveillance, robot navigation, image interpolation, or physical and metrological applications. Fischer et al. presented an end-to-end optical flow estimation network using CNNs and generated a synthetic dataset to train supervised learning-based networks [28]. Ilg et al. stacked the network to improve the overall quality and introduced a subnetwork specialized for small motions [46]. Other researchers [47,48] employed a pyramid down/up-sampling architecture to estimate the refined flows, especially in occluded regions. Recently, Teed et al. built multi-scale 3D correlation volumes for all pixels and iteratively updated a flow field using a recurrent unit to produce precise results for optical flow datasets and reasonable estimation of unseen real data [27]. We employed the network as the baseline of our system and fine-tuned its pre-trained weight to detect windshield wiper masks by providing a synthesized dataset.

#### 2.3.2. Optical Flow Datasets

Generating a precise pixel-wise ground-truth optical flow from real data is challenging because matching pixels between sequential images and obtaining displacement vectors is nearly impossible. For this reason, most optical flow datasets are composed of synthetic data, including FlyingChairs [28], FlyingThings3D [29], and MPI-Sintel [30]. Menze et al. provided optical flow data from real images by collecting highly dynamic scenes from the KITTI dataset and extracting disparity maps using LiDAR and CAD models to move 3D point clouds to obtain the flow ground truth [49]. To address this painstaking work, Sun et al. rendered 2D synthetic data with learnable hyperparameter control properties, and their results were comparable to or better than methods with existing datasets [50]. Motivated by this idea, we synthesized images and flows of wipers into a prior optical flow dataset to improve data diversity and leverage it to fine-tune a pre-trained model to precisely detect fast-moving objects.

## 3. Approach

### 3.1. System Overview

We propose behind-the-scenes (BTS), a wiper-occlusion canceling model that detects windshield wipers in captured images from an in-vehicle frontal view camera sensor while driving in adverse rain environments. As depicted in Figure 4, we acquired real image data by driving ourselves because of the limited data diversity of prior optical flow datasets. We annotated hundreds of ground truth wiper masks for evaluation using only a software tool. The pseudo-data were sampled and subsequently synthesized within Sintel to cause intentional occlusions. We employ a supervised learning-based optical flow network, RAFT [27], as our baseline to leverage its reasonable performance with respect to unseen real data and the facility of applying various fine-tuning schedules. We follow the same training/fine-tuning schedules as RAFT to demonstrate the effectiveness of our data synthesis approach. BTS outperforms in wiper mask and scene detection by +0.025 SSIM, +9.4%p F1-WMD, and +7.6%p F1-WSD in terms of the prior best-performing model of RAFT, achieving 0.962 SSIM, 91.6% F1-WMD, and 88.3% F1-WSD. Vision-based image restoration and object detection applications demonstrate the importance and extensibility of BTS by leveraging precise wiper masks to remove occlusions and enable detection.

### 3.2. Data Acquisition

#### 3.2.1. Hardware Setup

We mounted a camera (Stereolabs ZED2 [51]) in the cockpit of our autonomous vehicle (a commercial model of KIA Niro HEV). The vehicle has a customized battery system to supply a stable voltage to the camera to capture high-resolution videos in real-time. The camera was mounted at the approximate center of the cockpit and close to the windshield, as depicted in Figure 5, and the camera Euler angles (yaw, pitch, and roll) were set near zero initially by monitoring calibration data from its built-in IMU sensor to record the general frontal view. The videos were stored on a laptop (Acer Predator Triton 500) which the camera was connected.

#### 3.2.2. Recording Environment

We extracted images from the recorded videos and acquired more than 150 k images at a resolution of 1080×1920 with a 30-Hz frame rate under rainy weather conditions in two urban areas (Songdo and Dongchun) and a motorway located in Incheon Metropolitan City, South Korea. In addition, we collected images in different environments, including various locations, precipitation conditions, and seasons, for data diversity. As summarized in Figure 6, Songdo downtown contains many skyscrapers and broad roads, while Dongchun has more industrial infrastructure such as factories. The Incheon Airport motorway contains the broadest road with bridges, underpasses, and overpasses. Rain intensity (precipitation) is the most crucial factor for images because a higher intensity increases both adherent raindrops and wiper speeds that distort and occlude the images. In addition, a fast wiper is captured as a bent-shaped object.

According to the Korea Meteorological Administration reports, our dataset contains image data under daily precipitation totals of 12.4 mm (27 August 2020), 47.7 mm (19 November 2020), and 56.2 mm (3 April 2021), while the maximum hourly precipitation ranged from 9.0 mm/h to 13.5 mm/h. Our data were collected during the daytime between 09:00 and 16:00 in three different seasons: spring (April), summer (August), and fall (November). We divided the datasets into *WipersSpring*, *WipersSummer*, and *WipersFall* and exploited them for different purposes.

Pseudo-ground truth wiper data for training and ground truth wiper scene for the quantitative evaluation were generated from WipersSpring. Hand-crafted ground truth wiper data for the quantitative evaluation were generated from WipersSummer and test images were selected from WipersSummer and WipersFall.

#### 3.2.3. Hand-Crafted Ground Truth

The main purpose of the ground truth is only to evaluate our prediction results regarding the wiper mask and scene, so we annotated the corresponding ground truths. We manually classified 34,960 images into *true* or *false* for wiper scenes based on the existence of severe truncation or occlusion by windshield wipers, and 6029 images were labeled as wiper scenes in the WipersSpring dataset. The ground truth data were utilized to evaluate wiper scene detection (WSD). As depicted in Figure 7, we annotated wiper pixels of 215 images in WipersSummer using a software annotation tool, PixelAnnotation [52], with a graphical tablet, WACOM CTL-672, to extract accurate and precise ground truth for objective pixel-wise evaluation of wiper mask detection (WMD). Because the hand-crafted annotation process took approximately 1 h to generate five ground truth wiper masks, we adopted auto-labeling to generate pseudo-ground truth for data synthesis 100 times faster.

### 3.3. Data Synthesis

#### 3.3.1. Pseudo-Ground Truth

The pseudo-ground truth in data synthesis is leveraged to address the limited data diversity of typical optical flow datasets and improve the wiper detection accuracy efficiently. Prior optical flow datasets, such as FlyingChairs [28], FlyingThings3D [29], Sintel [30], and KITTI 2015 [49], contain static object movements and small occlusion regions, which are entirely different from the actual images with windshield wipers. Hence, adding the absent large fast-moving objects increases data diversity. However, obtaining ground truth flow vector information in real data is nearly impossible, and generating hand-crafted pixel-wise wiper masks is a time-consuming and inefficient task. Therefore, we applied auto-labeling to efficiently acquire pseudo-ground truth masks and flows of actual windshield wipers and synthesized the data into a prior dataset to improve the data diversity in a simple but robust method.

We initially searched for the optimized flow vector magnitude threshold for heuristically classifying wipers out of predicted optical flows in WipersSpring, and the optimized range was measured between 25.0 and 50.0 for an image size of 360×640. By applying the thresholds to pre-trained models, raft-things was selected as the most applicable model for auto-labeling based on qualitative analysis. We selected images with no objects existing for applying auto-labeling. Then, we heuristically classified 199 plausible data to the pseudo-ground truth wiper masks and flows among the inaccurate auto-labeled outputs, as shown in Figure 8. Even though the ground truth contained inaccurate pixel or flow estimation, we leveraged the data without any post-processing to demonstrate the effects of utilizing pseudo-data.

#### 3.3.2. Synthesis Scenario

Applying appropriate scenarios for sampling pseudo-ground truth for data synthesis is essential to our proposed method for enhancing data diversity. Unlike slow-moving objects in other typical optical flow datasets, Sintel is derived from a 3D animated short film composed of dynamic objects captured by a virtual shooting camera in 23 scenes of image sequences. Significantly, longer sequences are suitable for applying diverse sequential or random combinations of pseudo-data. Our idea is to mount virtual wipers to the virtual shooting camera lens to create similar occlusions by synthesizing the pseudo-ground truth wiper data into Sintel.

We built various synthesis scenarios based on the combinations of sequences, which were classified in terms of wiper appearances instead of applying the original sequences of wipers. The wipers generally appeared in three different temporal parts of the image sequences during a single wiper cycle, and we classified them as *starting (**S**)*, *returning (**R**)*, and *ending (**E**)* states, as illustrated in Figure 9a.

The starting state contains the most extended image sequence because of the relatively slow movement as the wipers begin operating upward from the initial resting state. The shapes of the wipers are less distorted because the camera shutter speed is sufficient to capture its original shape, and thus the occluding area is smaller than the others. However, after a few frames, the returning state includes images of more distorted shapes and larger occlusions as the wipers move downward faster than in the previous state. Finally, the most abnormal shapes are observed in the ending state because the wipers return to the initial point at a much faster speed, so the slower camera shutter speed catches a severely distorted shape.

Therefore, we leveraged the features of wipers in different states to create various data synthesis scenarios. Synthesizing wiper data sequences was implemented based on the idea of mimicking mounting a wiper onto a virtual shooting camera that moves independently, regardless of the original spatio-temporal features. In addition, we randomly injected single wiper data into other sequences to simulate unnatural wiper sequences for data diversity. As a result, as illustrated in Figure 9, we implemented five different synthesis scenarios in the original Sintel (Figure 9b). A *single sequence* (Figure 9c) synthesized a single wiper sequence to parts of the original scene to mimic one wiper cycle. In a *single ending* (Figure 9d), we randomly sampled a single ending state wiper image and synthesized one image among the scenes. A *sequential sequence* (Figure 9e) sampled sets of the sequential states and attached them to the entire scene, as in a typical wiper cycle. To ignore temporal features, the *random* (Figure 9f) scenario sampled sequence or an image randomly and synthesized all images in a scene, while *random sequence* (Figure 9g) randomly sampled an image sequence of the states to rearrange the order of temporal features. The actual samples of the sequential sequence and random scenarios are depicted in Figure 10.

## 4. Experiments

### 4.1. Implementation Details

We adopted the Pytorch implementation of supervised learning-based RAFT as the baseline of our system because other networks such as FlowNet 2.0 [46] and PWC-Net [47] cannot adapt to the unseen real domain data. Our final model, *BTS*, restored the weight of the pre-trained RAFT’s model, raft-things, which were trained with FlyingChairs and FlyingThings3D. We then fine-tuned the weight with our synthesized dataset, *SintelWipers*, which includes 2 × 1064 images (*clean* and *final*) and 1041 flows (*flow*) in the training dataset. The model was trained with two TITAN RTX GPUs based on the following training schedule and parameters, as summarized in Table 1. Furthermore, the vector magnitude threshold was set to 25.0, and the optimized value for the models was obtained from a threshold search. BTS was quantitatively and qualitatively evaluated in terms of models by comparing to pre-trained models of RAFT, which were trained with the typical datasets such as FlyingChairs, FlyingThings3D, Sintel, and KITTI 2015, and in terms of different data synthesis scenarios.

Images in WipersSummer and WipersFall were utilized for the quantitative and qualitative evaluations. For the quantitative evaluation, the structural similarity index measure (SSIM) [53] and binary classification (BC) were utilized to assess the image similarity and pixel-wise matching accuracy of the binary wiper mask detection (WMD) and accuracy of the wiper scene detection (WSD). However, our data are imbalanced because one object class occupies 17.1% (8,472,972 pixels) of the entire set. Therefore, the F1-score in binary classification is considered as the objective indicator because it is the harmonic mean of recall and precision calculated with the true positive (*TP*), false negative (*FN*), and false positive (*FP*) as follows:(1)$$Precision=\frac{TP}{TP+FP},Recall=\frac{TP}{TP+FN},F1=2\times \frac{Precision\times Recall}{Precision+Recall}$$

### 4.2. Quantitative Evaluation

BTS outperformed the other models in both SSIM and binary classification evaluations while producing 15 frames per second, as summarized in Table 2. An average SSIM of 0.962 and a standard deviation of 0.027 demonstrated the similarity of the overall results to the ground truth with respect to luminance, contrast, and structure. Furthermore, the highest harmonic mean F1-scores manifested how the model is balanced in precision and recall. By fine-tuning our method with the KITTI dataset with the same training schedule as raft-kitti, the fine-tuning results were significantly improved compared with raft-kitti.

Raft-chairs scored the worst and seemed unable to overcome the domain mismatch because the model was trained with unrealistic and steady-moving synthetic objects different from the target object. Instead, raft-things, trained with more complex and diverse 3D objects, made significant improvements of +10.4%p SSIM, +59.7%p F1-WMD, and +42.6%p F1-WSD by adapting to the real domain. Even though raft-sintel was trained more with dynamic, diverse, and sequential frames of synthetic data, the performance was degraded. In contrast, fine-tuning with driving environment-based KITTI deteriorated by −18%p F1-WMD. However, by replacing Sintel with our generated synthetic datasets for fine-tuning, SintelWiper made the model much more accurate and balanced, proven by the +10.8%p F1-WMD and +7.6%p F1-WSD compared with raft-things. Higher F1 scores were accomplished by significantly improving recall in WMD and WSD. This means that BTS detects the pixels of wipers that other models missed, especially wipers in the ending states, which can be compared intuitively in the qualitative results.

Moreover, the optical flow estimation EPE errors of BTS on the original datasets were slightly increased to less than +0.27%p, which is still an acceptable performance compared with the others, as summarized in Table 3. The experiment demonstrated that BTS is capable of manipulating the optical flow and wiper datasets.

As summarized in Table 4, we observed the influences of various sampling scenarios for data synthesis using combinations of the starting, returning, and ending states mentioned in Section 3.3.2. By comparing partial and complete proportions of synthesis, more data led to improved performance. Interestingly, a single sequence caused an adverse effect that worsened the performance compared with applying nothing to the original data. According to the results, sampling sequential frames was generally beneficial for improving recall and maintaining a balance precision of 88.1% F1-WMD. Random sampling predicted slightly more wiper pixels to score +1.1%p higher F1-WMD. Therefore, we leveraged sequential and random sampling combinations to achieve the best performance, with 0.926 SSIM, 91.6% F1-WMD, and 88.3% F1-WSD. The numerical gaps and effectiveness in terms of the various factors in different scenarios are intuitive in qualitative evaluations.

### 4.3. Qualitative Evaluation

The predicted wiper mask for each model was overlapped with the original image for an intuitive qualitative comparison, as depicted in Figure 11. As anticipated from the quantitative evaluation, raft-chairs showed unstable and inaccurate performance. Raft-kitti detected more parts of wipers than raft-chairs, but the model often classified the background as a wiper and vice versa. Raft-sintel and raft-things had minute differences in the numbers, but the visualized results showed noticeable differences in shape completion. Even though raft-sintel was trained with an additional synthetic dataset that included more dynamic and sequential images, the model failed to cope with the real domain, in contrast to raft-things.

BTS demonstrated outstanding performance regardless of the speed of the wipers, as categorized by the bounding boxes in Figure 11. In the starting state, raft-things, raft-sintel, and raft-kitti were detected plausibly when BTS tended to classify a narrow background region inside the wiper for more precise shape completion. The wipers in the returning state seemed tricky for other models to adapt to at faster speeds, whereas BTS was detected accurately. As mentioned, the fastest wiper appears as a severely distorted object with a large occlusion in the image. Although the intractable wiper deteriorated other models, BTS precisely detected the abnormal-shaped object and classified the background inside the wiper.

In Figure 12, we compare images of the ending state to demonstrate how the proposed data synthesis method significantly improved the prior models. The remarkable advancements are intuitively shown by comparing our results with those of raft-sintel, which proves the effectiveness of data diversity. The numerical gaps in the quantitative evaluations visualized the apparent performance gaps. Synthesizing wiper sequences improved the pre-trained models to adapt to the wipers in relatively slower movements, such as starting and returning states. Randomly sampling wiper sequences advanced the detection of the deteriorated shapes of wipers. Consequently, data synthesis by harmonizing sequential and random sampling of wiper sequences achieved precise and dominant performance.

### 4.4. Applications

#### 4.4.1. Image Restoration

Our main purpose of image restoration is to remove wiper-occlusions in images, restore information behind the wipers, and obtain clear visibility. We utilized a flow-based video completion application, FGVC [54], which requires masks of target regions as input data and leverages information of image sequences for restoring the regions. We employed binary masks generated by raft-things and BTS to compare the effectiveness of precise binary masks for assigning target regions to restore.

As illustrated in Figure 13a, wipers occluded meaningful pixels of objects such as vehicles, traffic signs, and road features in the scenes. The images are improper to be fed as input data into other vision-based algorithms to provide sufficient information due to the wiper-occlusions. Image restoration results utilizing raft-things generated artifacts (Figure 13b), while BTS produced comparable outputs (Figure 13d). Assigning accurate target masks for restoration of reconstructed building walls and signs, BTS contributed to avoiding artifacts, such as mirrored and blurred vehicle wheels, while it contributed to the network to produce clearer restored outputs by guiding precise region information. Consequently, leveraging BTS into image restoration enabled the utilization of all images with wiper-occlusion regions by visualizing objects, such as vehicles and road features that were previously behind-the-scenes. The restored images can be utilized as refined input data to other vision-based models for ADAS tasks, such as object detection.

#### 4.4.2. Object Detection

Object detection is one of the core applications in ADAS because it outputs bounding boxes for surrounding objects to provide core information regarding confidence, location, size, number, and existence. Therefore, erroneous detection results may cause inaccurate decisions by ADAS to endanger safety. We tested a representative vision-based 2D object detection network, YOLOv4 [24], using the restored images generated by the FGVC and BTS to demonstrate how the misjudgments of detection were improved.

Since it is impossible to obtain corresponding ground truth images and requires expensive efforts to annotate bounding boxes of objects due to wiper-occlusions, we presented the test images with truncated or occluded objects whose predicted bounding boxes were qualitatively inaccurate. As illustrated in Figure 14, the network struggled to detect truncated or occluded vehicles with the original images with wipers. Truncated objects were predicted with less confidence or separated into more than two bounding boxes, while occluded objects did not have a chance to be detected. However, after restoring the images with precise wiper masks, the truncated and occluded objects became visible and were detected accurately. The quantitative evaluation demonstrated improvements of object detection by 199.9% to original images and 31.2% to restored images with raft-things, as summarized in Table 5. These improvements enable enhancement in lateral and longitudinal controls such as ACC or AEB, thereby contributing to safer and more precise driving performances under rainy weather conditions by removing truncations or occlusions by wipers in image input data.

## 5. Discussion

We achieved the detection of vehicle windshield wipers driving under rainy weather conditions, leveraging a synthesized optical flow dataset with generated pseudo-ground truth wiper data by auto-labeling acquired real datasets. Although the proposed method exploits optical flow for pixel-wise object detection, the brightness constancy assumption, which can deteriorate the optical flow functionality, was not considered. Empirically, the brightness under rainy weather conditions is stable compared with that under clear weather conditions because of the absence of the sun. Therefore, we tested our method, which was fine-tuned with rainy data only, to unseen images captured on a sunny day where fatal backlighting occurred. Figure 15 shows that our method is robust enough to detect wipers in brighter weather and backlit situations. If an image is either too bright or dark, software-manner image recovery, such as deep learning-based high dynamic range (HDR) [55] can resolve the visibility to reduce potential errors.

Even though our datasets recorded ordinary driving circumstances for hours to capture various truncations or occlusions, there were no abnormal cases. However, it could be a concern when a wiper completely occludes an object in sequential frames. If our method provides such images to image restoration as inputs, the images would be recovered as if the object never existed. Assuming that such a situation occurred in the frames in Figure 14, where an object moving from right to left through two lanes with a width of 2.75 m each was occluded completely for three sequential frames, its velocity can be calculated using the following parameters: frame rate (*f*), number of frames (*n*), and distance (*d*).
(2)$$v=\frac{f\times d}{n}=\frac{30\;\mathrm{Hz}\times (\frac{2.75\;m}{\mathrm{lane}}\times 2\mathrm{lanes})}{3frames}=\frac{55\;m}{s}=198\;\mathrm{km}/h$$

The object should move at 198 km/h (approximately 123.8 mph) in the image through the calculation, which is uncommon and illegal in most driving environments so the authors believe that complete occlusions for sequential frames rarely occur. However, it would be a limitation of BTS if the case of a fast-moving object at the speed appears.

Although the proposed method accommodates minor errors in the input images, limitations exist under heavy rain in the night-time. Higher hourly precipitation increases the number of adherent raindrops on a windshield that merge into larger droplets before the wipers operate. As a result, severe image deterioration occurs as a result of unpredictable light refraction and reflection in the droplets caused by countless random light sources in the darkness, including traffic lights, vehicle head/rear lamps, and street lamps. The authors believe that the complicated features of objects is one reason why classical hardware is still applied to vehicles in the 21st century. Resolving the complexity adherent raindrops for removal is our next step to overcome heavy rain without existence of wipers.

The proposed method focuses on simple, efficient, and robust real-time wiper detection for driving under rainy weather conditions. As illustrated in the application demonstration in Section 4.4, our method plays an essential role by proving compatibility, utility, ability, and extensibility to other vision-based tasks commonly used in ADAS in vehicles for pursuing safety such as panoptic segmentation [56], as depicted in Figure 16. We can expand data diversity by obtaining clear frontal views in datasets that incorporate driving under rainy conditions by restoring wiper-occluding images instead of skipping wiper frames.

In addition, we expect to leverage our model to auto-label pseudo-ground truth for panoptic segmentation to observe capabilities and to apply various software techniques, such as pre-/post-processing, to develop our method in future work to search for a breakthrough. Developing the detection of unintentional occluding objects for vision-based tasks in various industries, including surveillance, anomaly detection, autonomous vehicles, and unmanned aerial vehicles (UAV), offers promising future applications for the proposed method.

## 6. Conclusions

In this study, we developed a novel real-time windshield wiper-occlusion canceling model, BTS, for driving under adverse rain conditions by leveraging optical flow for data synthesis with pseudo-ground truth using auto-labeling. The purpose of the proposed technique is to provide a deep learning-based optical flow model to provide a precise mask for a fast-moving wiper that occurs in massive occlusion of image data. We acquired 150 k real images of driving in rainy environments in a metropolitan city in South Korea to augment a prior dataset for fine-tuning and to obtain hand-crafted ground truth wiper masks and scenes for objective evaluation. We synthesized the fast-moving object onto synthetic image sequences as if a virtual wiper was mounted on a virtual camera. We achieved accurate pixel-wise wiper mask detection by scoring an average of 0.962 (+0.025) with a standard deviation of 0.027 (−0.050) in SSIM, and an F1-score of 91.6% (+9.4%p) in binary classification, while wiper image scene detection showed an F1-score of 88.3% (+7.6%p). BTS demonstrated significant enhancements by applying it to vision-based applications to provide precise occluding regions for generating wiper-free images to enable object detection on previously invisible images. The proposed method also proved the compatibility, utility, ability, and extensibility to other vision-based applications can be leveraged to enhance their performance under adverse rain environments.

## Figures and Tables

**Figure 1 sensors-21-08081-f001:**
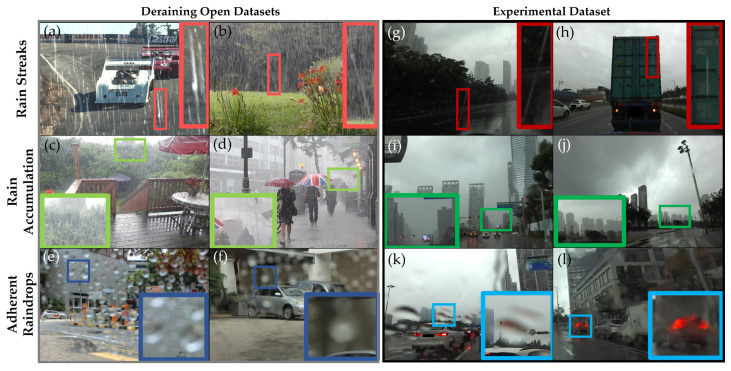
Difference in rain effects between open datasets and experimental dataset: (**a**–**f**) deraining open datasets [10,11,15]; (**g**–**l**) our experimental dataset. Rain streaks (1st column) are mostly invisible in the images in the experimental data compared with the open dataset. Rain accumulation (2nd column) occurs at a far distance with no specific impacts on nearer objects. Instead, both artificial and actual adherent raindrops (3rd column) distort the images and deteriorate visibilities. Best viewed with zoom-in.

**Figure 2 sensors-21-08081-f002:**
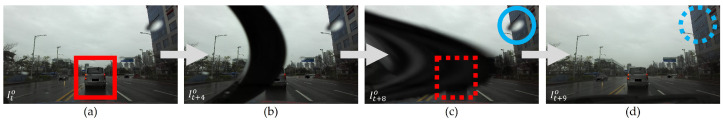
Problematic wiper effects in an experimental image sequence for time frames: (**a**) the frontal vehicle (solid red) is visible; (**b**) the wiper truncates the vehicle; (**c**) the wiper completely occludes the vehicle (dashed red) to wipe the adherent raindrop (solid blue); (**d**) the vehicle is back to visible, and the adherent raindrop is cleared (dashed blue). Best viewed with zoom-in.

**Figure 3 sensors-21-08081-f003:**
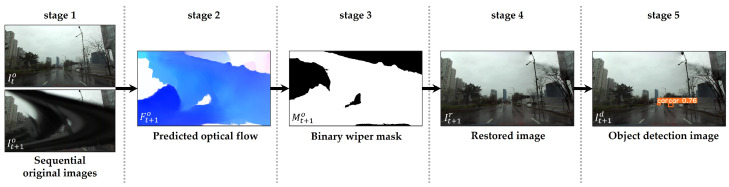
Processing pipeline of inference task: (**stage 1**) takes two sequential input images $\left(I_{t}^{o},I_{t+1}^{o}\right)$; (**stage 2**) predicts optical flow $\left(F_{t+1}^{o}\right)$; (**stage 3**) extracts binary wiper mask $\left(M_{t+1}^{o}\right)$; (**stage 4**) restores the wiper-occlusion region in the image $\left(I_{t+1}^{r}\right)$; (**stage 5**) outputs object detection image $\left(I_{t+1}^{d}\right)$. Best viewed with zoom-in.

**Figure 4 sensors-21-08081-f004:**
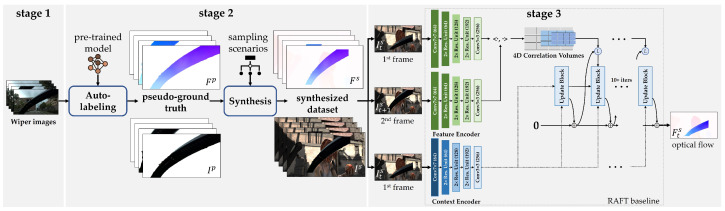
Training pipeline of BTS: (**stage 1**) data acquisition: we acquired real image data under rainy weather conditions, including windshield wipers; (**stage 2**) data synthesis: pseudo-ground truth wiper mask and flow data are generated from the collected data using auto-labeling, after which we conduct data synthesis to improve the data diversity of typical optical flow data; (**stage 3**) training: RAFT baseline [27], a supervised learning-based optical flow network, is adopted to fine-tune its pre-trained models with our synthesized dataset, SintelWipers.

**Figure 5 sensors-21-08081-f005:**
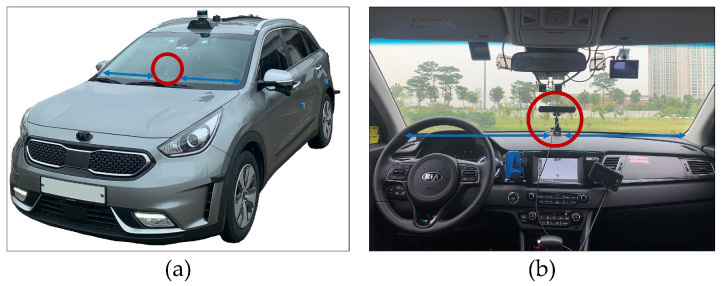
Hardware setup for data acquisition: (**a**) experimental vehicle; (**b**) the camera in the cockpit to capture the frontal view of driving under rainy weather conditions.

**Figure 6 sensors-21-08081-f006:**
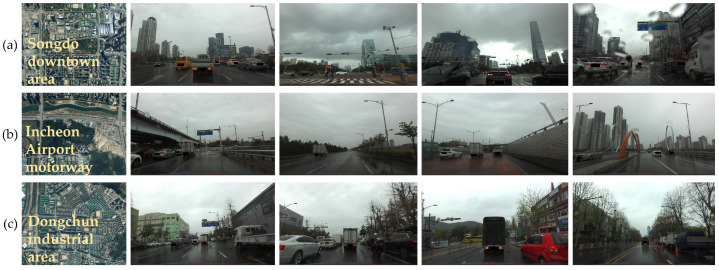
Recording environments: (**a**) Songdo downtown area; (**b**) Incheon Airport motorway; (**c**) Dongchun industrial area. Satellite images (1st column) and sample images (2nd–5th columns) of the three areas show the infrastructure differences.

**Figure 7 sensors-21-08081-f007:**
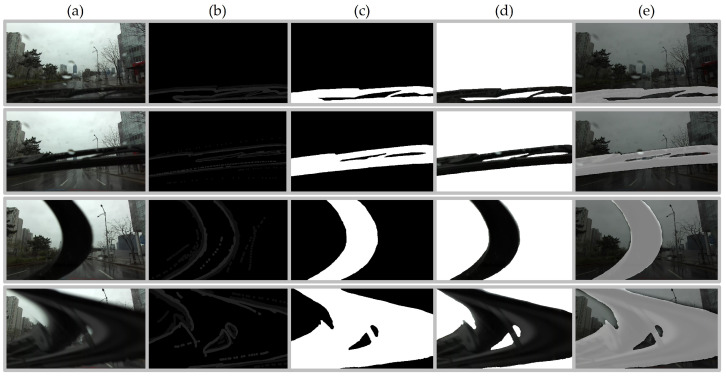
Hand-crafted ground truth wiper mask generation: (**a**) original image; (**b**) manual markings; generated (**c**) binary mask; (**d**) RGB mask; (**e**) concatenated original image (**a**) and binary mask (**c**). Best viewed with zoom-in. Songdo downtown area; Incheon Airport motorway; Dongchun industrial area. Satellite images (1st column) and sample images (2nd–5th columns) of the three areas show the infrastructure differences. Best viewed with zoom-in.

**Figure 8 sensors-21-08081-f008:**
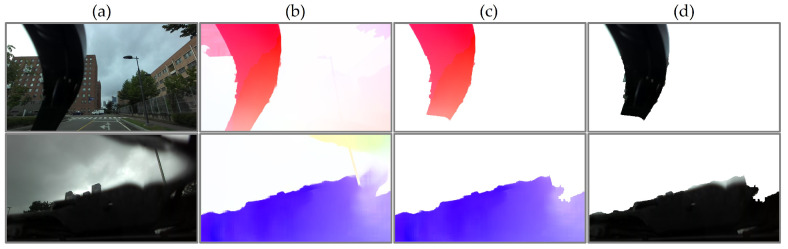
Pseudo-ground truth generation: (**a**) original images; (**b**) predicted optical flow by auto-labeling; (**c**) threshold filtered flow; (**d**) extracted RGB masks.

**Figure 9 sensors-21-08081-f009:**
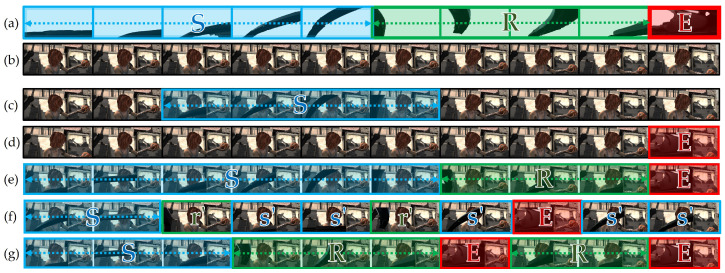
Data synthesis scenario examples: (**a**) original wiper image sequence; (**b**) original Sintel [30]; synthesized with the scenario of (**c**) *single sequence*; (**d**) *single ending*; (**e**) *sequential sequence*; (**f**) *random*; (**g**) *random sequence*. A general wiper image sequence can be divided into starting, returning, and ending states according to the wiper movement. For convenience, we denote each state’s image sequence set as a capital letter (*S*, *R*, and *E*) and a single wiper image as a small letter with apostrophe (*s’*, *r’*, and *e’*). For instance, the synthesized image sequence of (f) is composed of 1 × (starting image set) + 1 × (returning image) + 2 × (starting image) + 1 × (returning image) + 1 × (starting image) + 1 × (ending image set)+ 2 × (starting image). Each row includes ten sequential image samples to show the differences between synthesis scenarios in the original Sintel. Best viewed with zoom-in.

**Figure 10 sensors-21-08081-f010:**
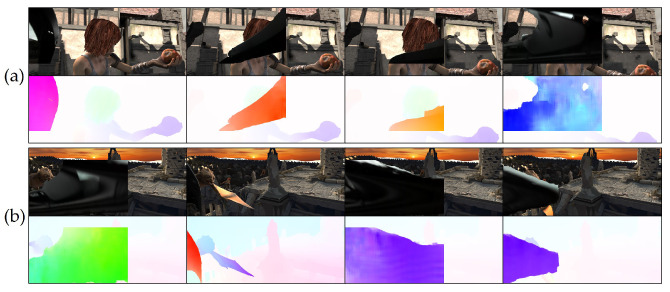
Examples of actual synthesized images and flows in Sintel [30] with scenarios: (**a**) *sequential sequence* synthesized image sequences of returning and ending states sequentially; (**b**) *random* synthesized random single images among the states.

**Figure 11 sensors-21-08081-f011:**
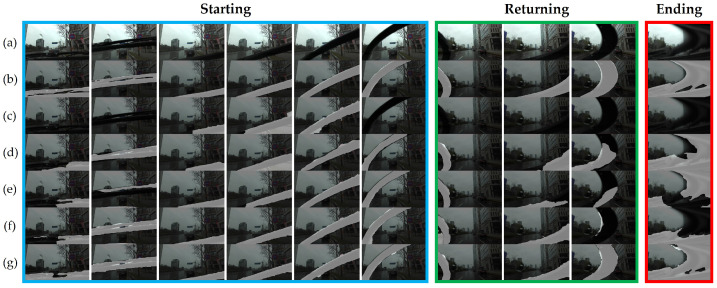
Qualitative results in terms of model: (**a**) original image; (**b**) ground truth wiper masks; wiper detection results by (**c**) raft-chairs; (**d**) raft-things; (**e**) raft-sintel; (**f**) raft-kitti; (**g**) proposed BTS. Our methods demonstrates significant improvement compared with the performance of pre-trained models of RAFT. Starting state (blue), returning state (green), and ending state (red) are categorized by bounding boxes. Best viewed with zoom-in.

**Figure 12 sensors-21-08081-f012:**
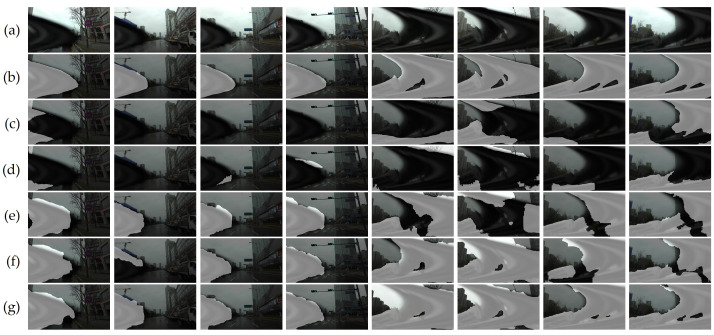
Qualitative results in data synthesis scenario: (**a**) original images; (**b**) ground truth wiper masks; wiper detection results by (**c**) raft-sintel; (**d**) raft-things; fine-tuned with scenario of (**e**) sequential sequence scenario; (**f**) random; (**g**) random sequence (BTS). (**c**,**d**) models were trained with the original Sintel dataset, while (**e**–**g**) models fine-tuned the reference model, raft-things, with various synthesis scenarios. BTS copes with the intractable severely distorted object resulting from the fast speeds of wipers. The results show the effectiveness of data diversity by our data synthesis strategies. Best viewed with zoom-in.

**Figure 13 sensors-21-08081-f013:**
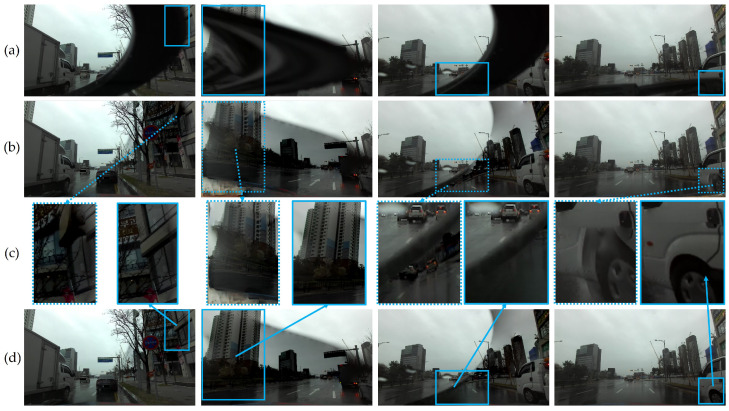
Image restoration results: (**a**) original images; (**b**) restored results with masks from raft-things; (**c**) comparisons between raft-things and BTS; (**d**) restored results with masks from BTS. The results demonstrate the importance of mask accuracy by reducing artifacts in the restored regions in (**c**), such as distorted buildings (1st and 2nd row), mirrored objects (3rd row), and a blurred vehicle wheel (4th row), as shown in the red boxes. Best viewed with zoom-in.

**Figure 14 sensors-21-08081-f014:**
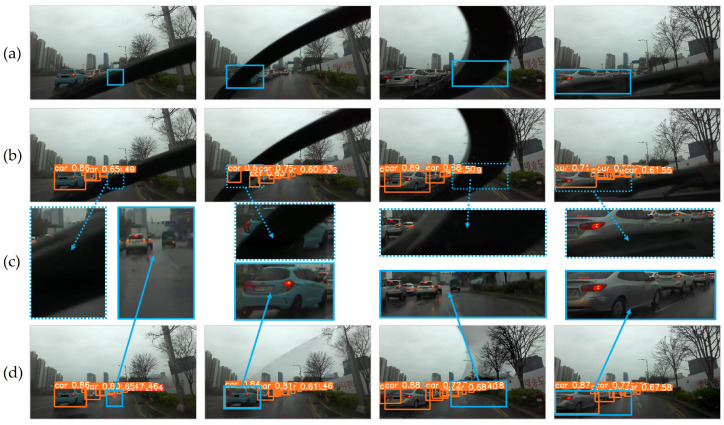
Object detection results: (**a**) original image sequence; (**b**) object detection before image restoration; (**c**) result comparisons; (**d**) object detection after image restoration. The wiper truncates (blue dashed box) or occludes (green dashed box) frontal objects, which causes erroneous object detection results regarding prediction properties of objects. Best viewed with zoom-in.

**Figure 15 sensors-21-08081-f015:**
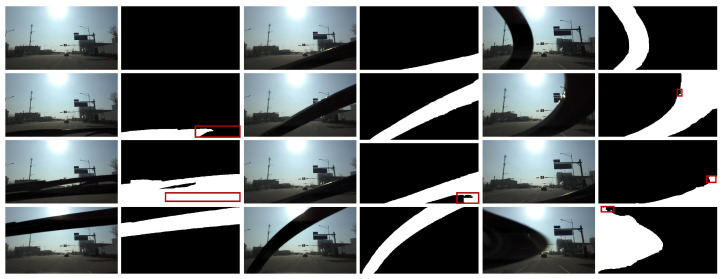
Wiper detection results in backlit situations under clear weather conditions: original wiper images and corresponding detected wiper masks (white area). Despite the solid backlit situation in clear weather conditions, our model accommodates adverse brightness conditions and shows plausible detection results. Best viewed with zoom-in.

**Figure 16 sensors-21-08081-f016:**
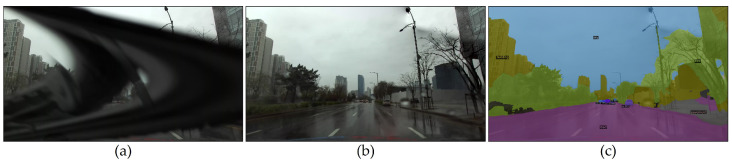
Panoptic segmentation result with leveraging BTS: (**a**) original wiper-occluded image; (**b**) restored image with a wiper mask from BTS; (**c**) panoptic segmentation result. BTS demonstrates the compatibility with other vision-based tasks, such as panoptic segmentation.

**Table 1 sensors-21-08081-t001:** Training Schedule for BTS.

Schedule	Dataset	#Iterations	Batch Size	Crop Size	Learning Rate	Weight Decay
1	FlyingChairs	100 k	12	386 × 496	0.0004	0.0001
2	FlyingThings3D	100 k	6	400 × 720	0.000125	0.0001
3	SintelWipers	100 k	6	368 × 768	0.000125	0.00001

**Table 2 sensors-21-08081-t002:** Experimental results of SSIM and binary classification (wiper mask and scene detection) in terms of models.

		Wiper Mask Detection (WMD)					Wiper Scene Detection (WSD)		
Model	Dataset	SSIM		Binary Classification			Binary Classification		
		Average	Std. Dev.	Precision	Recall	F1 Score	Precision	Recall	F1 Score
raft-chairs	C	0.833	0.112	63.4	13.7	22.5	74.5	25.6	38.1
raft-things	C + T	0.937	0.087	88.3	76.9	82.2	76.8	85.0	80.7
raft-sintel	C + T + S	0.934	0.094	92.8	71.6	80.8	73.6	84.7	78.8
raft-kitti	C + T + S/K	0.884	0.079	75.7	53.7	62.8	68.9	73.0	70.9
BTS	C + T + Sw	**0.962**	**0.027**	87.6	**96.0**	**91.6**	**87.4**	**89.2**	**88.3**
BTS-kitti	C + T + Sw/K	0.890	0.075	68.8	79.4	72.5	85.8	84.1	84.9
*Datasets: FlyingChairs (C), FlyingThings3D (T), Sintel (S), KITTI (K), SintelWipers (Sw)*									

**Table 3 sensors-21-08081-t003:** Experimental results of end-point-error (EPE) on original optical flow datasets.

Model	Sintel (Train)		KITTI 2015 (Train)	
	Clean	Final	F1-Epe	F1-All
raft-chairs	2.24	4.51	9.85	37.6
raft-things	1.46	2.78	5.00	17.4
raft-sintel	0.75	1.22	1.21	5.6
raft-kitti	4.55	6.15	0.63	1.5
BTS	**0.93**	**1.49**	4.37	13.5
BTS-kitti	5.41	6.68	**0.67**	**1.7**

**Table 4 sensors-21-08081-t004:** Results of raft-things fine-tuned with various data synthesis scenarios.

		Wiper Mask Detection (WMD)					Wiper Scene Detection (WSD)		
Proportion	Method	SSIM		Binary Classification			Binary Classification		
		Average	Std. Dev.	Precision	Recall	F1	Precision	Recall	F1
	Orig. Sintel	0.934	0.094	92.8	71.6	80.8	73.6	84.7	78.8
Partial	Single seq.	0.922	0.095	93.4	62.9	75.2	71.8	76.5	74.1
	Single end.	0.938	0.070	0.924	74.1	82.2	95.2	76.0	84.5
	Sequential	0.953	0.045	0.886	87.7	88.1	90.8	80.1	85.1
Complete	Random	0.956	0.032	0.879	90.5	89.2	91.3	82.6	86.7
	**Rand. Seq.**	**0.962**	**0.027**	**0.876**	**96.0**	**91.6**	**87.4**	**89.2**	**88.3**
*Reference model: raft-sintel (original Sintel), raft-things + synthesized Sintel with scenarios*									

**Table 5 sensors-21-08081-t005:** Quantitative results: average precision (AP) comparison of object detection.

Image Type	Model for Mask Generation	Average Precision (AP, %) @ IoU = 0.5
original	none	23.30
restored	raft-things	53.26
	**BTS**	**69.87**

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
