# Peer review of "Behind-The-Scenes (BTS): Wiper-Occlusion Canceling for Advanced Driver Assistance Systems in Adverse Rain Environments"

_sensors, 2021, doi:10.3390/s21238081_

Round 1

Reviewer 1 Report

Thank you for the paper. It treats a relevant topic that can improve existing systems. However, there are some weaknesses through the manuscript which need improvement. My comments and suggestions are as follows:

  1. Abstract gives information on the main feature of the performed study, but some details about why it is needed to perform the analysis should be added as well as future applications.
  2. In the introduction, indicators such as accidents that can be avoided based on real data or previous analysis could show its relevance.
  3. As the practical relevance has been clearly stated, authors must clarify necessity of the performed research. Objectives of the study also differences with the previous researches must be clearly mentioned in the last part of introduction.
  4. Also, an analysis of scientific and practical gaps regarding the functionality developed in the paper needs to be presented considering if it has been treated or analyzed in other applications or sectors.
  5. Before or after showing figure 3, it has to be explained how the restored version is generated. How the wiper occlusion is “filled” i.e. it is a simple copy of the I (0, t) or it has a logic behind to forecast the events and based on it build the restored image.
  6. The literature study of previous approaches presented in chapter 2 must be classified according to predefined criteria: input uncertainty, data or image treatment, benefits and disadvantages of the solution, interface with drivers, car and other objects, etc.
  7. It would be nice, if authors could add a syndication of how the data is acquired, when (frequency), where it is stored, etc. A table or figure with an overview can help to show the procedure followed.
  8. Following the previous comment, 3.3. needs to provide a better explanation of which of the data acquired has been treated, with which method with the justification of its selection based on the research goal, as well as with a description of the parametrization of the method including other characteristics.
  9. Chapter 4 needs to be extended with more detail description of methodological steps as well as characteristics such as datasets used and models in comparison.
  10. Chapter 4 also needs to define in advance which are the target criteria for a comparison between models in different scenarios.
  11. The sub-chapter 4.4. needs to be associated with the models presented otherwise should be included in the discussion section. The logic behind the restoration and object detection should be also included.
  12. In the discussion section, although 198km/h is uncommon or illegal, there are non-limit highways as well as rallies in which this speed is reached, so it should be included as a comment and limitation.
  13. The discussion section can be improved by adding other applications in the same or other sectors using the same approach as well as adding a scheme of the advantages and disadvantages of the system.
  14. The conclusion needs to be improved in order to change from a conclusion as a towards a conclusion including limitations and future research options.

Author Response

First of all, we appreciate the valuable comments from the learned reviewer.

We attached our responses to your comments.

Thank you.

Reviewer 2 Report

The Authors present a novel real-time windshield wiper-occlusion canceling model BTS (behind-the-scenes). The method is based on using optical flow to classify fast-moving wiper flows. The structural similarity index measure (SSIM) and binary classification metrics were applied to evaluate the wiper mask and scene detection.

The paper is interesting, correctly organized and well written. Results of exhaustive experiments  with a specially prepared real dataset of driving under adverse rainy weather conditions show the usefulness of the approach and are convincing.

Remarks:

Please remark that the difference between two percentages is termed as percentage point, not %.

Please make your dataset publicly available.

Author Response

(The authors gave the same response as above.)

Reviewer 3 Report

This is a very nice and original paper, as most works in the literature focus on the cancellation of rain effects but not on the cancellation of wiper appearance.

I think that the presentation of the proposed work can be improved in the following two ways:

  • The first is to provide a supplementary video that demonstrates the result of the method. The fact the proposed method is important for cancelling the visual motion by wipers calls for a video example.
  • The second is to explain why cancelling wiper motion is an advantage over competing methods. In other words, if methods exist that make the need of wipers obsolete, then why is a method that cancels wipers is needed?

Author Response

(The authors gave the same response as above.)

Reviewer 4 Report

The authors propose behind-the-scenes (BTS), a wiper-occlusion canceling model that detects windshield wipers in captured images from an in-vehicle frontal view camera sensor while driving in adverse rain environments. They employ a supervised learning-based optical flow network, RAFT, as their baseline to leverage its reasonable performance with respect to unseen real data and the facility of applying various fine-tuning schedules. BTS demonstrated enhancements by applying it to vision-based applications to provide precise occluding regions for generating wiper-free images to enable object detection on previously invisible images. 

Following are my observations to improve the manuscript.

[1] It is thought that the ultimate purpose of this study is to improve the road object detection performance by restoring a clean image by removing wipers, rain streaks, and raindrops in a raining environment. Besides the wiper problem, how can you suggest a solution to the rain streaks and raindrops problem?

[2] Can you present the quantitative evaluation results of the BTS algorithm in the object detection application described in Section 4.4.2?

[3] Using an in-vehicle network, ADAS cameras can receive wiper operation status via CAN message. It is likely that you can suggest a more improved wiper removal algorithm using the received wiper operation status.

Author Response

(The authors gave the same response as above.)

Round 2

Reviewer 1 Report

Dear Authors, 

Although the paper has significantly improved, it is recommended to review the following points before final acceptance:

1. Improve the clarity of the text, figures and data in chapters 3 and 4.

2. Sub-chapter 3.3. needs to provide a better explanation of  which of the data acquired has been treated, with which method with the justification of its  selection based on the research goal, as well as with a description of the parameterization of the method including other characteristics. A graphical representation or a table can help to depict or classify the information.

3. Chapter 4 needs to be extended with more detail description of methodological steps as well as characteristics such as datasets used and models in comparison. A figure with the methodological steps can improve the explanation of the research methodology.

Best regards

Author Response

We thank the valuable observations from the learned reviewer.

We attached our responses to your comments.

Thank you

Reviewer 3 Report

I think that the authors have sufficiently responded to all the reviewers' comments.

Author Response

Dear learned reviewer,

We could improve the paper according to your great suggestions.

We appreciate your reviews and efforts to enhance our study.

Thank you very much.

Best regards,

Junekyo Jhung.